# Are early childhood stunting and catch-up growth associated with school age cognition? —Evidence from an Indian birth cohort

**Beena Koshy**[1]*, **Manikandan Srinivasan**[2], **Sowmiya Gopalakrishnan**[1], **Venkata Raghava Mohan**[3], **Rebecca Scharf**[4], **Laura Murray-Kolb**[5], **Sushil John**[6], **Rachel Beulah**[1], **Jayaprakash Muliyil**[2], **Gagandeep Kang**[2]

**1** Developmental Paediatrics Unit, Christian Medical College, Vellore, India, **2** Wellcome research Unit, Christian Medical College, Vellore, India, **3** Community Health, Christian Medical College, Vellore, India, **4** University of Virginia Children's Hospital, Charlottesville, Virginia, United States of America, **5** The Pennsylvania State University, State College, Pennsylvania, United States of America, **6** Low Cost Effective Care Unit, Christian Medical College, Vellore, India

* beenakurien@cmcvellore.ac.in

**Data Availability Statement:** All relevant data are within the Supporting Information file.

## Abstract

### Background

Millions of children worldwide especially in the Asian subcontinent are vulnerable to early childhood stunting. There are contradictory reports of the association between catch-up growth in childhood and school age cognition.

### Methods

A community-based birth cohort recruited between 2010 and 2012 from urban slums in Vellore, India was followed up until 9 years of age. From regular anthropometric measurements, stunting status for each individual child was calculated at 2, 5 and 9 years. Cognition was assessed at 9 years of age using the Malin's Intelligence Scale for Indian Children (MISIC). Children were divided into groups based on stunting at each time point as well as catch-up growth, and a regression model was utilised to evaluate their association with cognition at 9 years.

### Results

Among 203 children included in this analysis, 94/203 (46.31%) children were stunted at 2 years of age, of whom 39.36% had a catch-up growth at 5 years of age, and 38.30% at 9 years. Around 10% of the cohort remained stunted at all time points. In the multivariable analysis, children who were stunted at 2, 5 and 9 years had a significantly lower verbal and total intelligence quotient (IQ) scores by 4.6 points compared to those who were never stunted. Children with catch up growth following stunting at 2 years had higher cognition scores than those who were persistently stunted throughout the childhood.

### Conclusions

This study showed persistent stunting in childhood was associated with lowering of 4–5 IQ points in childhood cognition at 9 years of age. Recovery from early life stunting in children

**Funding:** 1. The Etiology, Risk Factors and Interactions of Enteric Infections and Malnutrition and the Consequence for Child Health and Development Project (MAL-ED) is carried out as a collaborative project supported by the Bill and Melinda Gates Foundation, the Foundation for the NIH and the National Institutes of Health/Fogarty International Center (Grant number – GR-681) BK, LMK, RS, VRM, SJ, GK - GK lead Sponsor/Funder had no role in the study design, data collection and analysis, decision to publish or preparation of manuscript. 2. The 9-year follow-up of the Mal-ed India cohort was supported by an Intermediate clinical and public health research fellowship awarded by the DBT/Wellcome Trust India Alliance to Dr. BK. (Fellowship grant number IA/CPHI/19/1/ 504611). Sponsor/Funder had no role in the study design, data collection and analysis, decision to publish or preparation of manuscript.

**Competing interests:** No competing interests exist

with catch up growth prevented further lowering of cognition scores in these children compared to persistently stunted children. Nutritional supplementation during late infancy and early toddlerhood in addition to continuing nutritional supplementation programmes for preschool and school children can improve childhood stunting and cognitive abilities in vulnerable populations.

## Introduction

The sustainable development goal (SDG) 4 of achieving quality education by 2030 [1] is vulnerable to ongoing early childhood risks. Twin risks of early childhood stunting, and absolute poverty have caused over 200 million young children worldwide, majority of them in the low- and middle-income countries (LMIC), to have suboptimal developmental potential [2–4]. Around 150 million under-five children globally are stunted, defined as height for age z scores (HAZ) < -2 Standard Deviation (SD) on WHO growth charts [5], with more than half belonging to the Asian continent [6]. A sub-national analysis of child malnutrition in India found average child stunting to be 39.3%, with states having low socio-demographic index of development being disproportionately more affected [7]. Children globally have similar growth potential in life, as evidenced by comparable growth performance of affluent Indian children to that in developed countries [8]; but those growing up in vulnerable environments are exposed to early childhood risks affecting not just their linear growth, but also developmental potential.

The first 1000 days of life are critical for optimal developmental potential as brain development happens during this period through neurogenesis, neuronal migration, axonal and dendritic growth, synaptogenesis, myelination, and synaptic pruning [9], and any disruption to this neuronal process can affect long-term structural and functional capacity of the brain [2]. Children exposed to early childhood risk factors thus have not just stunting, but also developmental, cognitive and learning difficulties. Stunting has been identified as a major public health priority due to its association with an individual's morbidity, mortality, and reduced developmental, learning and economic potential, and its propagation of 'intergenerational cycle of poverty' [3, 10]. Meta analyses have shown consistent relationship between early childhood stunting with child cognitive and motor development [11] and later adult height, but an inconclusive association with adult educational potential and achievement [12].

Human growth and development are continuous processes throughout the whole lifespan, and periods beyond the first 1000 days of life are also vital to optimise an individual's potential [13]. Longitudinal studies have shown that catch-up growth can happen between 24 months and mid-childhood and during adolescence years [14]. There are contradictory findings in the literature on the extent of cognition recovery due to catch-up in linear growth, with some studies reporting no association with cognition or development [15–17], while others reporting catch-up growth even in late childhood improving cognitive deficits in stunted children [18, 19]. These contradictory findings may be due to differing definitions and time periods studied [20, 21]. Absolute catch-up is defined as a child growing up to match the mean healthy reference sample height to narrow the height-for-age difference (HAD), while a relative catch-up is an improvement in the HAZ score [20, 21]. Catch-up growth can be also defined by stunting status at each time point where children once stunted grow up to attain HAZ scores within normal limits (≥ -2 SD) in subsequent follow-up [18, 22–24].

In this context, birth cohort studies, especially in LMIC settings such as India, can help in our understanding of the pattern of undernutrition and recovery from birth throughout the

childhood, and its impact on cognition in the childhood. Birth cohort studies can also compare different growth trajectories through childhood and thus providing an opportunity to plan for timing of interventions to improve overall growth capacity and in turn economic and human potential. The objective of the current study was to evaluate the association of stunting and catch-up growth with childhood cognition at 9 years in an urban slum birth cohort setting in India.

## Materials and methods

### Study population and methodology

The current study population was a birth cohort originally enrolled for the 'Etiology, Risk Factors and Interactions of Enteric Infections and Malnutrition and the Consequences for Child Health and Development' (MAL-ED) Network, a longitudinal prospective multinational, cohort study conducted in eight countries across the world [25]. The Indian study site was in Vellore, South India and consisted of eight adjacent densely populated urban slum dwellings [26]. Details about study population, recruitment process, exclusion criteria and follow-up are already published [26–29].

The initial birth cohort recruitment was conducted between March 2010 and February 2012 and recruited 251 new-borns. Children were subsequently followed up at 2, 3, 5, 7 and 9 years of age. The original birth cohort recruitment and subsequent follow-ups were approved by the Institutional Review Board and Ethics committee of Christian Medical College Vellore, and children were recruited at each stage after written informed parental consent with additional child assent at 9 years of age.

### Measures

Length of children was measured to the nearest cm using infantometer until 2 years of age. Stadiometer was used for further height measurements after 2 years of age. Weight was measured using an electronic weighing scale to the nearest 10 grams. Head circumference was measured using a non-stretchable tape by a trained person to the nearest 0.1 cm. All machines underwent periodic recalibration, and personnel had periodic retraining as per the MAL-ED protocol. For follow-ups separate training was conducted for all personnel involved. To calculate z-scores for height/weight for age, Multicentre Growth Reference Study (MGRS) standards was used for measurements done under-5 years and WHO AnthroPlus software for measurements performed at 9 years [5]. Stunting was defined as HAZ < -2 on WHO growth curves [5] and catch-up in growth defined as HAZ within normal SDs on WHO charts after previous instance/s of stunting [20]. Current definition used for catch-up growth using stunting status at each time points has been widely used in other cohort studies from similar LMIC settings for growth trajectory modelling [15, 21–24]. Advantage of use of this definition based on HAZ scores, would be that HAZ is a relative measure accounting for variability in growth in the reference population, compared to measures based on HAD, which is an absolute measure for growth.

**The Malin's Intelligence Scale for Indian Children (MISIC).** The Malin's Intelligence Scale for Indian Children (MISIC) [30] is the Indian adaptation of the Wechsler Intelligence Scale for Children (WISC). This test can be administered to children from 6 years to 16 years.

Verbal sub-scales measure the range of factual information in terms of vocabulary, and the ability to comprehend and reason by logical and deductive thinking. The verbal scale has subtests of Information, Similarities, Arithmetic, Vocabulary, Comprehension and Digit Span. Performance sub-scales measure visual perception, spatial organization and co-ordination. The performance scale has subtests of Picture Completion, Coding, Picture Arrangement, Block Design and Object Assembly. This measure was administered in a distraction-free

environment in the community clinic setting by a single psychologist. Raw scores were converted into standardized scores and corresponding quotients were computed using mental and chronological ages. Verbal intelligence quotient (VIQ) was derived from verbal subscales, performance intelligence quotient (PIQ) from performance sub-scales and total intelligence quotient (IQ) using all subscales.

**The WAMI measure for socio-economic status.** Socio-economic status (SES) was measured in the MAL-ED study using a composite measure with components of access to improved **W**ater and sanitation, **A**ssets, **M**aternal education and total household **I**ncome (WAMI) [31]. A trained field worker visited the home and administered the translated and piloted WAMI measure.

**Raven's progressive matrices.** The Raven's progressive matrices measures non-verbal reasoning ability and can be used as culture-fair measure [32]. A single trained psychologist administered this measure to all mothers at 6–8 months of child's age, as per the MAL-ED study protocol, to assess maternal cognition [33]. We used raw scores for our analysis.

### Data entry and analysis

Completed paper forms were validated by field supervisor before data entry. Information from the paper forms were entered into electronic database using a double entry database system managed by the Data Co-ordinating Centre of the MAL-ED study [25].

### Statistical analysis

Categorical variables including sociodemographic characteristics and anthropometric measurements were expressed as percentages and compared across the time-points between birth and 9 years of age. Families of children with WAMI scores less than 33rd percentile were classified as low for socioeconomic status and those who scored 33rd percentile and above were classified as high.

Based on their stunting status at 2, 5 and 9 years of age, children were grouped into four categories–i) children who were never stunted, ii) children stunted at 2 years with a catch-up at 5 years, iii) children stunted at 2 and 5 years, with a catch-up at 9 years, and iv) children stunted at 2, 5 and 9 years. Continuous variables such as mother's cognition, cognitive assessment scores in children using MISIC scale were expressed as mean and standard deviation. Statistical significance of verbal IQ, performance IQ and total IQ between the groups was tested using Analysis of variance (ANOVA). Simple linear regression analysis was used to model the association of stunting on cognition scores analysed at 9 years. Having the children who were never stunted as the reference category, effect of stunting at one or more time points on cognition scores, under verbal and performance domain, was assessed separately after adjusting for mother's cognition and socioeconomic status at 2 years of age in the multivariable model. Beta co-efficients along with 95% confidence interval were reported. Model fit was assessed using $R^2$ values and p-value $< 0.05$ was considered as the level of significance. Stata version 13 (StataCorp. 2013. Stata Statistical Software. Release 13. College Station, TX: StataCorp LP) software was used for the statistical analysis.

### Results

Between 2010 and 2012, 301 pregnant mothers residing in the study area were screened for recruiting 251 new-borns in the original birth-cohort. Details about study population, recruitment, exclusion criteria and follow-up are already published [26–29, 34]. There were 228 (90.84%), 212 (84.46%) and 205 (81.67%) children available for follow-up at 2, 5 and 9 years respectively (Fig 1). Families moving out of the study area was documented as the

predominant reason for loss to follow-up in the study and this finding from the same birth-cohort has already been published [29]. The 9-year recruitment was planned between February 2019 and February 2021. Due to the Covid-19 pandemic, recruitment was paused for > 6 months in 2020. Anthropometric and cognitive assessments for all children were completed by April 2021.

This birth-cohort was established in urban settlements of Vellore and families of children in this cohort had their adults working as predominantly unskilled labourers. Access to safe water and sanitation facility was sub-optimal for the study families and 17% of babies weighed less than 2.5 kg at birth. Cohort characteristics at enrolment, 2, 5 and 9 years were similar with respect to sex distribution and SES characteristics (Table 1). Mothers' mean cognition raw score (SD) was 43.91 (10.49) on Raven's matrices. Proportion of children who were stunted (101/228–44.5%) and under-weight (81/228–35.7%) was highest by 2 years of age and this declined by 5 years and more so by 9 years (Table 1 and Fig 2).

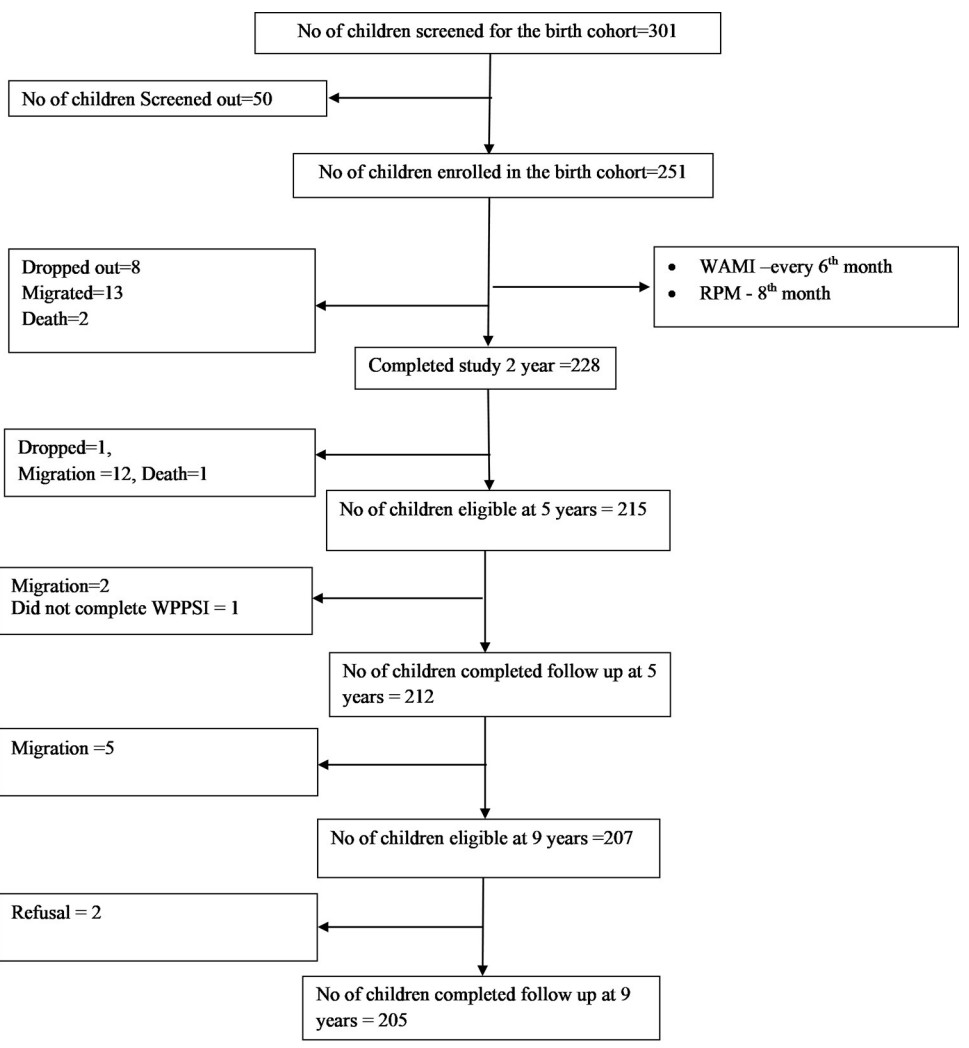

WAMI – Water and sanitation, Assets, Maternal education and Income score for socio-economic status; RPM – Raven's progressive matrices

**Fig 1. Flowchart depicting the follow-up of the birth cohort.**

**Table 1. Comparison of cohort characteristics at birth enrolment, years 2, 5 and 9 of follow-up of MAL-ED cohort.**

| | Enrolment (At birth) n = 251 (%) | 2 years n = 228[*] (%) | 5 years n = 212 (%) | 9 years n = 205[#] (%) |
|---|---|---|---|---|
| **Gender** | | | | |
| Male | 113 (45) | 105 (46.05) | 98 (46.23) | 96 (46.83) |
| Female | 138 (55) | 123 (53.95) | 114 (53.77) | 109 (53.17) |
| **Socioeconomic status (SES)** | | | | |
| Low *(WAMI < 33rd percentile)* | 71 (30.2) | 71 (31.14) | 65 (30.66) | - |
| High *(WAMI ≥ 33rd percentile)* | 164 (69.79) | 157 (68.86) | 147 (69.34) | |
| **Height-for-age Z scores (HAZ)** | | | | |
| ≥ - 2 SD | 210 (83.67) | 126 (55.51) | 150 (70.75) | 183 (89.27) |
| < - 2 to ≥ - 3 SD | 31 (12.35) | 69 (30.40) | 50 (23.58) | 20 (9.76) |
| < -3 SD | 10 (3.98) | 32 (14.10) | 12 (5.66) | 2 (0.98) |
| **Weight-for-age Z scores (WAZ)** | | | | |
| ≥ - 2 SD | 194 (77.29) | 146 (64.32) | 149 (70.28) | 146 (73) |
| < - 2 to ≥ - 3 SD | 41 (16.33) | 61 (26.87) | 52 (24.53) | 34 (17) |
| < -3 SD | 16 (6.37) | 20 (8.81) | 11 (5.19) | 20 (10) |

[*]Only 227 children had information on WAZ and HAZ scores at 2 years of follow-up.

[#]Only 205 children had information on WAZ scores at 9 years of follow-up.

MAL-ED—Etiology, Risk Factors and Interactions of Enteric Infections and Malnutrition and the Consequences for Child Health and Development; IQ–Intelligence Quotient.

About 203/251 children in the cohort had complete information on stunting at all time points. However, to study the association between early childhood stunting with/without a catch-up at later ages and cognition at 9 years, only 200/203 were considered for analysis. Remaining 3/203 children were not included in the analysis, as 2/3 children were stunted only at 5 years and the other child only at 9 years, with normal growth parameters at other time-points. 91/203 children were stunted at 2 years of age, of whom 34/91 (37.36%) had a catch-up

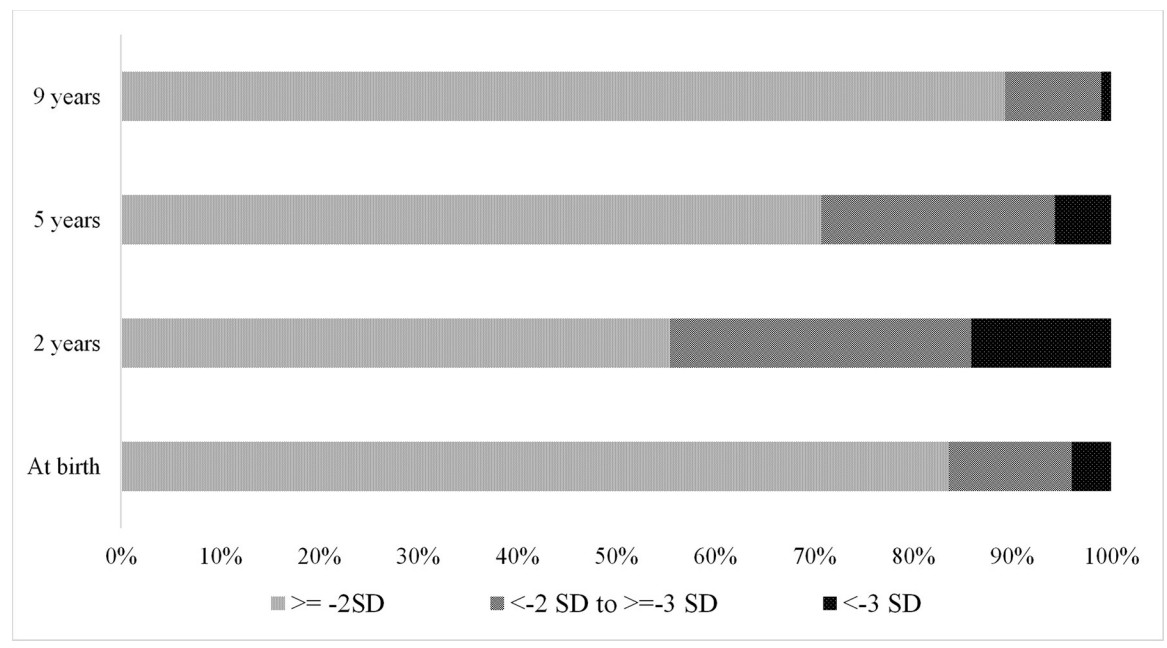

**Fig 2. Proportion of children stunted at 0, 2, 5 and 9 years of age in MAL-ED cohort.**

**Table 2. Grouping of children in MAL-ED cohort based on stunting* status at 2, 5 and 9 years (n = 200#).**

| Groups | n (%) | Mean (SD) HAZ scores at birth | Mean (SD) HAZ scores at 2 years | Mean (SD) HAZ scores at 5 years | Mean (SD) HAZ scores at 9 years |
|---|---|---|---|---|---|
| Children who were never stunted | 109 (54.50) | -0.65 (0.91) | -1.20 (0.58) | -0.93 (0.63) | -0.26 (0.76) |
| Children stunted at 2 years and catch-up at 5 years | 34 (17) | -1.20 (1.05) | -2.30 (0.21) | -1.52 (0.34) | -0.79 (0.46) |
| Children stunted at 2 and 5 years, and, catch-up at 9 years | 36 (18) | -1.32 (0.84) | -2.88 (0.56) | -2.47 (0.36) | -1.06 (1.87) |
| Children stunted at 2, 5 and 9 years | 21 (10.50) | -1.67 (1.3) | -3.32 (0.75) | -3.00 (0.46) | -2.48 (0.38) |

Children with height-for-age z scores < - 2SD were classified as stunted.

#Of 203 children who had complete information on stunting at all three time points, 200 (98.52%) were included in one of the above groups. The remaining 3/203 (1.48%) children who were not included in any of the above groups, two were stunted only at 5 years, with their HAZ scores at 2 and 9 years ≥ 2 SD, and, another child was stunted only at 9 years, followed by HAZ scores ≥ 2 SD at 2 and 5 years.

growth at 5 years of age, 36/91 (39.56%) remained stunted at 5 years and had a catch-up growth by 9 years, and 21/91 (23.08%) continued to remain stunted at 5 and 9 years (Table 2) The mean HAZ score of the cohort measured at birth, 2, 5 and 9 years of age was -0.97, -1.89, -1.55 and -0.73 respectively. Children in all groups showed the lowest HAZ score at 2 years of age with subsequent improvements at 5 and 9 years of age.

The mean VIQ, PIQ and total IQ (SD) scores of children at 9 years of age were 94.03 (9.71), 91.49 (13.01) and 92.76 (10.26) respectively. Distribution of cognition scores across the groups based on stunting status of children at one or more time points is presented in Table 3. Children who were never stunted scored high in VIQ and total IQ assessment with mean (SD) scores of 95.19 (9.51) and 94.37 (9.75) respectively, compared to those who were stunted at one or more time points. In the multivariable analysis (Table 4), after adjusting for mother's cognition scores and socioeconomic status at 2 years, children who were stunted at 2, 5 and 9 years had significantly lower verbal IQ scores by 4.6 points lesser than those who were never stunted (VIQ—89.15 vs 95.19 respectively; adjusted beta co-efficient (95% CI): -4.61 (-8.86 - -0.36)). In performance domain, children who were stunted at 2 years, but recovered later had lesser IQ scores by 5.8 points when compared to those of who were never stunted during the follow-up (PIQ– 87.88 vs 93.54 respectively; adjusted beta co-efficient (95% CI): -5.77 (-10.51 -

**Table 3. Distribution of verbal, performance and total cognition raw scores, and IQ at 9 years in children who were stunted at various time points of follow-up in MAL-ED cohort (N = 200#).**

| Groups, n | Verbal scores, mean (SD) | | | Performance scores, mean (SD) | | | Total scores, mean (SD) | | |
|---|---|---|---|---|---|---|---|---|---|
| | Raw scores | IQ | P value* | Raw scores | IQ | P value* | Raw scores | IQ | P value* |
| Children who were never stunted (n = 109) | 475.96 (47.56) | 95.19 (9.51) | 0.024 | 467.64 (61.87) | 93.54 (12.40) | 0.051 | 943.60 (97.48) | 94.37 (9.75) | 0.036 |
| Stunted at 2 years and catch-up at 5 years (n = 34) | 474.18 (47.93) | 94.85 (9.54) | | 439.41 (83.67) | 87.88 (16.73) | | 913.59 (122.65) | 91.36 (12.27) | |
| Stunted at 2 and 5 years, and, catch-up at 9 years (n = 36) | 457.75 (49.88) | 91.55 (9.98) | | 452.36 (59.68) | 90.47 (11.94) | | 910.11 (101.13) | 91.01 (10.11) | |
| Stunted at 2, 5 and 9 years (n = 21) | 445.76 (45.77) | 89.15 (9.15) | | 435.79 (51.51) | 87.16 (10.30) | | 881.55 (84.86) | 88.16 (8.49) | |

#200/203(98.52%) children who had complete information on stunting at all three time points were included in this analysis.

*Statistical significance between the groups for verbal IQ, performance IQ and total IQ was tested using ANOVA.

**Table 4. Association of stunting status at 2, 5 and 9 years and verbal, performance and total IQ scores at 9 years of age using regression analysis (n = 200#).**

| Predictors | Adjusted beta co-efficient (95% CI) | | |
|---|---|---|---|
| | **Verbal IQ** | **Performance IQ** | **Total IQ** |
| **Stunting** | | | |
| *Children who were never stunted* | 1 | 1 | 1 |
| *Stunted at 2 years and catch-up at 5 years* | -0.20 (-3.70–3.29) | **-5.77 (-10.51 - -1.03)** | -3.00 (-6.65–0.66) |
| *Stunted at 2 and 5 years, and, catch-up at 9 years* | -1.79 (-5.27–1.69) | -0.88 (-5.60–3.85) | -1.33 (-4.97–2.31) |
| *Stunted at 2, 5 and 9 years* | **-4.61 (-8.86 - -0.36)** | -4.64 (-10.41–1.14) | **-4.62 (-9.07 - -0.17)** |
| **Socio economic status at 2 years** | | | |
| *Low (<33rd percentile of WAMI)* | 1 | 1 | 1 |
| *High (> = 33rd percentile of WAMI)* | **5.13 (2.27–7.98)** | **4.91 (1.04–8.79)** | **5.02 (2.04–8.01)** |
| **Mother's IQ scores** | **0.18 (0.06–0.31)** | **0.33 (0.16–0.49)** | **0.26 (0.13–0.38)** |

IQ–Intelligence Quotient; WAMI–Water, Assets, Maternal Education and Income

#200/203(98.52%) children who had complete information on stunting at all three time points were included in this analysis.

Bolded values have significant p-values

$R^2$ values for regression models with following outcomes -verbal, performance and total IQ scores were 14.88%, 13.96% and 17.06% respectively.

-1.03)). $R^2$ values for models based on verbal and performance outcome was 14.88% and 13.96% respectively. Children who were stunted at 2, 5 and 9 years had significantly lesser total IQ scores by 4.6 points compared to those who were never stunted (IQ– 88.16 Vs 94.37; adjusted beta co-efficient (95% CI): -4.62 (-9.07 - -0.17)).

## Discussion

The present study evaluated childhood catch-up growth after early childhood stunting and its association with cognition at 9 years of age in a LMIC urban slum setting in India. Proportion of stunting in cohort children was noted to be the highest at 2 years of age with subsequent catch-up growth at 5 and 9 years of age. Children who were never stunted had better intelligence by 4–5 IQ points than those stunted throughout childhood. Children who were stunted at 2 years and had a catch up later had higher cognition scores compared to persistently stunted children. For verbal cognition, a graded response was seen where children whose height improved earlier performed better than those who caught up later.

In this birth cohort study, maximum stunting was observed at 2 years of age, where more than 40% of the cohort children were stunted. All 5 birth cohorts in the 'Consortium on Health Orientated Research in Transitional Societies' (COHORT) study was noted to have a similar nadir at 2 years of age [35]. In the current study, more than 75% of children who were stunted at 2 years caught up by 9 years of age, with only 10% of the original cohort being stunted at all time points. This is similar to another study published from Kerala where 50% children who were stunted at 0–4 years of age caught up by 7–11 years [36], but the percentage of children who caught up in linear growth in the current study is higher. Our finding is in discordance with the COHORT study follow-up, where longitudinal growth catch-up was noted by childhood in other birth cohorts, but not in the Indian site [37]. However, the HAZ at 2 years in the Indian COHORT study site was -1.9 [35], which is similar to what is reported in the current study. The COHORT study was conducted in New Delhi in north India in the 1980s and the Kerala study the beginning of 2010–20 decade. The present study has followed children for 9 years in the 2010–20 decade, spilling over to the current one. It is possible that national

nutrition intervention schemes for children through preschool programmes such as Balwadis and school midday noon meal schemes are better implemented currently to provide the catch-up growth as seen in this study. Balwadi program initiated in 1970–71 by the Government of India caters to health, nutritional and educational needs of preschool children aged 3 years onwards, while midday meal schemes support one wholesome meal for school going children [38, 39]. It is also imperative that such nutritional support schemes should continue for school going children as evidence from the Young Lives study show that linear growth faltering and catch-up also happen in late childhood and adolescent years between 8 and 15 years of age [18].

Early childhood stunting has been shown to be associated with childhood cognitive and motor abilities [11]. The Benin Demographic and Health Survey (BDHS) in Benin, Africa [40], Indian, Ghana and Peruvian studies [41–43] and the MAL-ED study conducted across 8 countries [44] have shown detrimental association of early childhood stunting on early childhood development in preschool years itself. The Cebu Longitudinal Health and Nutrition Study in Philippines [23, 45], the Pelotas birth cohort in Brazil, Birth to 20 plus cohort in South Africa [46, 47] and studies in Peru [48] and Thailand [49] provided evidence of early stunting being associated with poor later childhood cognition. The COHORT study done in Brazil, Guatemala, India, Philippines, and South Africa [50] and the Pelotas birth cohort study in Brazil [47] have shown that early stunting was associated with poor educational achievement in adulthood, though a subsequent meta-analysis showed inconclusive evidence [12]. A unit increase in HAZ at < 2 years of age resulted in 0.22 SD increase in cognitive ability in the childhood (5–11 years) in another meta-analysis [11]. The current study concurs with the existing literature that persistent childhood stunting is associated with lower verbal, performance and full cognition scores in mid-childhood and shows a lower cognition score of 0.3 SD (4–5 IQ points) in children who were persistently stunted. Stunting represents chronic undernourishment during early years of life in children, when rapid neurodevelopmental process takes place. It has been documented in human as well as animal studies that undernourishment during early childhood affects the neurodevelopment process such as neuronal growth, synaptogenesis and myelination process. In addition to undernutrition, literature has shown that there is an interplay of factors such as micronutrients intake, child's interaction with external environment, caregiver's behaviour, and timing and recovery from undernutrition, which holistically determine cognitive development process in children [51, 52]. Common and intersecting risks and causative pathways can affect both child growth and cognition, but stunting is generally recognised as the best surrogate for overall child health especially in resource poor settings [10].

Differing definitions of linear growth catch-up and time points utilised have given conflicting reports of its benefit in the literature [20]. As 2 years is the culmination of the first 1000 days of life including the gestational period, utilising this time point is more appropriate as done in the current analysis. The Young Lives study done in Ethiopia, India, Peru, and Vietnam showed that the catch-up growth between 1 and 8 years of age was associated with better receptive vocabulary, reading comprehension and mathematic achievement [53] and between 8 and 15 years was associated with better cognition and schooling outcomes in adolescence [18]. However, a study done in north India reported no relationship between linear growth change from early to late childhood and cognition in late childhood [54], but this study recruited children initially at an age range between 6–30 months. Another Indian study evaluating catch-up growth showed that school-level nutritional interventions caused better growth, but not the corresponding cognitive ability [17], probably due to an earlier onset insult to brain growth and cognition, as well as the short nature of intervention (6 months). Contrastingly, a rural Malawian study reported that late childhood growth was related to better cognition during the adolescence [55]. However, the present study showed that stunted children

with catch-up growth between 2 and 9 years of age had relatively higher cognition scores than the persistently stunted children, highlighting that earlier recovery after toddlerhood stunting prevented further lowering of cognition scores in these children compared to those who were persistently stunted. Thus, catch-up in growth in this period did not overcome all cognitive deficits due to early childhood stunting and is concurrent with other published literature [19, 23]. Another finding from our study that children who were stunted at 2 years and recovered by 5 years had a significantly lower performance IQ scores compared to children who were never stunted, highlights the fact that interventions to prevent early life stunting are not substitutable. Thus, our study adds to the evidence that consequence of early deprivation is not fully reversible, and it is important to identify nutritional deficiencies, in terms of calorie, protein and micronutrient requirements during the first 1000 days of life in children with appropriate, corrective interventions at the earliest, so as to maximize the gain in human capital.

Limitations of the current study include comparatively smaller sample size and some minimal loss to follow-up due to migration of families outside the study area. Strengths of the MAL-ED India cohort are strong range of data availability for early childhood, good follow-up rates at 2, 5 and 9 years of the cohort, standardised assessments with good quality control, and India-specific cognitive assessment in childhood.

## Conclusions

The current study showed persistent stunting in childhood was associated with a 0.3 SD lowering in childhood cognition attainment at 9 years of age and an early catch-up before 5 years prevented further lowering of cognition scores, although the cognitive deficits as a result of early nutritional deprivation persisted. Continuing nutritional supplementation programmes for preschool and school children can help in improving overall growth parameters as well as optimal education as envisaged in the SDG 4 [1]. Additional nutritional supplementation during late infancy and early toddlerhood can improve the stunting nadir seen at 2 years of age, and this has to be complemented with policy measures improving overall socioeconomic status of the family and improving educational attainment among mothers. Extending Balwadi or community support to this group with provision of home delivery of nutritional supplements might help this cause.

## Supporting information

**S1 File.**
(XLS)

## Acknowledgments

The authors thank the participants, their families and staff of the MAL-ED Network project.

## Author Contributions

**Conceptualization:** Beena Koshy, Rebecca Scharf, Laura Murray-Kolb, Jayaprakash Muliyil, Gagandeep Kang.

**Data curation:** Beena Koshy, Manikandan Srinivasan, Sowmiya Gopalakrishnan, Venkata Raghava Mohan.

**Formal analysis:** Manikandan Srinivasan, Venkata Raghava Mohan.

**Funding acquisition:** Beena Koshy, Gagandeep Kang.

**Methodology:** Beena Koshy, Venkata Raghava Mohan, Rebecca Scharf, Laura Murray-Kolb, Rachel Beulah.

**Project administration:** Beena Koshy, Venkata Raghava Mohan.

**Resources:** Beena Koshy.

**Supervision:** Beena Koshy, Sowmiya Gopalakrishnan, Sushil John, Rachel Beulah.

**Validation:** Rachel Beulah, Jayaprakash Muliyil.

**Writing – original draft:** Beena Koshy.

**Writing – review & editing:** Manikandan Srinivasan, Sowmiya Gopalakrishnan, Venkata Raghava Mohan, Rebecca Scharf, Laura Murray-Kolb, Sushil John, Rachel Beulah, Jayaprakash Muliyil, Gagandeep Kang.

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
