## [Decision Letter · Decision Letter 0]

27 Sep 2021

PONE-D-21-21418Are early childhood stunting and catch-up growth associated with school age cognition? – Evidence from an Indian birth cohort

PLOS ONE

Dear Dr. Koshy,

Thank you for submitting your manuscript to PLOS ONE. After careful consideration, we feel that it has merit but does not fully meet PLOS ONE’s publication criteria as it currently stands. Therefore, we invite you to submit a revised version of the manuscript that addresses the points raised during the review process.

ACADEMIC EDITOR:   

Thanks for sharing an interesting research with potentially a very useful addition to the literature.  The reviewers request to address some critical points as follows and discuss these in depth by reviewing the literature.

Additionally, regarding line 137-138, “These contradictory findings may be due to differing definitions and time periods studied [20],” although the authors discussed it in the discussion, you are requested to address in the introduction in terms of what your study add to the literature.

Also the potential to recover from undernutrition to neurocognitive development need to be addressed.

In Table 1, “Weight-for-age Z scores (HAZ)” needs to be changed to “Weight-for-age Z scores (WAZ)”

We look forward to receiving your revised manuscript.

Kind regards,

Seo Ah Hong, PhD

Academic Editor

PLOS ONE

Journal Requirements:

2. Thank you for stating the following in the Acknowledgments/ Financial Support Section of your manuscript: 

a. The Etiology, Risk Factors and Interactions of Enteric Infections and Malnutrition and the Consequence for Child Health and Development Project (MAL-ED) is carried out as a collaborative project supported by the Bill and Melinda Gates Foundation, the Foundation for the NIH and the National Institutes of Health/Fogarty International Center (Grant number – GR-681)

b. The 9-year follow-up of the Mal-ed India cohort was supported by an Intermediate clinical and public health research fellowship awarded by the DBT/Wellcome Trust India Alliance to Dr. BK. (Fellowship grant number IA/CPHI/19/1/504611)

Sponsors/Funders had no role in the study design, data collection and analysis, decision to publish or preparation of manuscript.

a. The Etiology, Risk Factors and Interactions of Enteric Infections and Malnutrition and the Consequence for Child Health and Development Project (MAL-ED) is carried out as a collaborative project supported by the Bill and Melinda Gates Foundation, the Foundation for the NIH and the National Institutes of Health/Fogarty International Center (Grant number – GR-681)

BK, LMK, RS, VRM, SJ, GK - GK lead

Sponsor/Funder had no role in the study design, data collection and analysis, decision to publish or preparation of manuscript.

b. The 9-year follow-up of the Mal-ed India cohort was supported by an Intermediate clinical and public health research fellowship awarded by the DBT/Wellcome Trust India Alliance to Dr. BK. (Fellowship grant number IA/CPHI/19/1/504611).

Sponsor/Funder had no role in the study design, data collection and analysis, decision to publish or preparation of manuscript.

No competing interests exist

Reviewers' comments:

Reviewer's Responses to Questions

**Comments to the Author**

1. Is the manuscript technically sound, and do the data support the conclusions?

Reviewer #1: Partly

Reviewer #2: Yes

2. Has the statistical analysis been performed appropriately and rigorously? 

Reviewer #1: No

Reviewer #2: Yes

3. Have the authors made all data underlying the findings in their manuscript fully available?

Reviewer #1: No

Reviewer #2: No

4. Is the manuscript presented in an intelligible fashion and written in standard English?

Reviewer #1: Yes

Reviewer #2: Yes

5. Review Comments to the Author

Reviewer #1: This is an interesting paper and is potentially a very useful addition to the literature. The cohort, while small, has a high level of retention and a number of informative measures. It is great, for example, that there is data on maternal cognition.

My concern with this paper is that the authors have not critically engaged with some of the key issues in the literature, despite mentioning a number, and these have significant implications for the interpretation of their results.

-The authors note that the definition of catch up is a point of dispute in the literature, but do not explain why they selected the one that they used. The one used is a rather weak definition - children can be defined as having caught up even when falling behind in terms of CM deficits. Nonetheless, it is a commonly used definition and its use could be justified, but it does need to be justified and not used without question.

-Stunting does not cause cognitive impairment, impairment and stunting are both indicators of malnutrition and possibly of lack of stimulation (and possible the combination of the two). As a result, any association between them needs to be carefully interpreted. The authors do not adequately cover this topic, which hinders the subsequent discussion of the results.

-If the authors are going to make the case that their results imply that nutrition programmes are needed, then a fuller discussion of the adequacy of the controls in the regression is needed. As mentioned, the association with stunting may be a result of smaller children being protected by parents leading to under stimulation - this may not be resolved by a nutrition program. Similarly, smaller children may be held back from school entry as parents do not think they are ready. The association between stunting and cognitive performance may be a result of less exposure to schooling.

-Do the authors think that catch up growth could indicate a recovery from earlier malnutrition leading to a similar catch up in cognition? Or that those who catch up avoid falling further behind, while those who remain stunted continue to fall behind? Their results could support either argument, but which argument is correct has important policy implications and this should be discussed.

The authors from time to time slip into causal language, this should be avoided.

The discussion on the importance of the first 1,000 days is a little unclear and could be improved. It is the pace and foundational nature of development in that period which is critical and this does not come through enough.

Overall the paper is well written, but there are a few places which would benefit from a careful edit.

Minor comments:

Why report weight if you don't use it? Just for context?

The top third based on WAMI can hardly be called high SES given the selection process. Would relatively high not be better?

When was the WAMI measure taken? What the implications of only using one time point? Did those who exhibited catch up also see an improvement in their SES which might explain the better cognitive scores?

Reviewer #2: Thank you for a very interesting study. Overall, the study is well conducted. There are a few comments and editorial as below.

1. Table 4: on the PIQ, it is shown that stunting at 2 y, though with a catch-up at 5 y had 5.77 PIQ less than normal children, but the children who stunted at 2 & 5 who only caught-up at 9 had much better PIQ than normal, than children who were already catch-up since 2y and those persisted till 9 y. What could explain such a pattern? Please add discussion.

2. Lines 365-366 & reference 17: Did children who received intervention were during older childhood (e.g., early school-age) and hence, the damage on cognition could have occurred since early age, aside from possible short duration of intervention? Please consider.

3. Editorial errors:

a. Line 116, Introduction: ‘…. (HAZ) <2 on WHO…’, should this read ‘…. (HAZ) <-2 SD…’?

b. Line 351 ‘A unit increase in HAZ <2 Y…’, suggest to read, ‘A unit increase in HAZ at <2 y of age..’

6. PLOS authors have the option to publish the peer review history of their article (what does this mean?). If published, this will include your full peer review and any attached files.

Reviewer #1: No

Reviewer #2: **Yes: **Pattanee Winichagoon

---

## [Author Response · Author response to Decision Letter 0]

12 Nov 2021

KINDLY SEE THE ATTACHED FILE 

To

The reviewers

PLOS One

Re: PONE-D-21-21418 - Are early childhood stunting and catch-up growth associated with school age cognition? – Evidence from an Indian birth cohort

Dear reviewers, 

Thank you for reviewing our manuscript titled “Are early childhood stunting and catch-up growth associated with school age cognition? – Evidence from an Indian birth cohort”. Author’s responses are provided below

Comments Authors’ response Page number; line number

Academic Editor 

1. Additionally, regarding line 137-138, “These contradictory findings may be due to differing definitions and time periods studied [20],” although the authors discussed it in the discussion, you are requested to address in the introduction in terms of what your study add to the literature. Thank you. Suggested change has been made.

 6; line 142-147

2. Also the potential to recover from undernutrition to neurocognitive development need to be addressed.

 Refer to the response to comment no. 1 6; line 142-147

3. In Table 1, “Weight-for-age Z scores (HAZ)” needs to be changed to “Weight-for-age Z scores (WAZ)”

 Thank you. Changed to WAZ Table 1

4. Financial statement Deleted from Title page section Cover letter

5. Competing interest No competing interest

Deleted from Title page section Cover letter

6. Data Minimal anonymized data set is uploaded Supplementary file

7. Ethics statement – Mention only in manuscript Deleted from Title page section 

Reviewer 1 

8. The authors note that the definition of catch up is a point of dispute in the literature, but do not explain why they selected the one that they used. The one used is a rather weak definition - children can be defined as having caught up even when falling behind in terms of CM deficits. Nonetheless, it is a commonly used definition and its use could be justified, but it does need to be justified and not used without question. Thank you. Suggested changes have been incorporated and corresponding references have been added. 7, Lines 175-178

9. -Stunting does not cause cognitive impairment, impairment and stunting are both indicators of malnutrition and possibly of lack of stimulation (and possible the combination of the two). As a result, any association between them needs to be carefully interpreted. The authors do not adequately cover this topic, which hinders the subsequent discussion of the results.

 Thank you. Suggested changes have been incorporated and corresponding references have been added. 19, Lines 365-372

10. If the authors are going to make the case that their results imply that nutrition programmes are needed, then a fuller discussion of the adequacy of the controls in the regression is needed. As mentioned, the association with stunting may be a result of smaller children being protected by parents leading to under stimulation - this may not be resolved by a nutrition program. Similarly, smaller children may be held back from school entry as parents do not think they are ready. The association between stunting and cognitive performance may be a result of less exposure to schooling. Thank you. Our analysis showed that apart from early childhood stunting, other factors such as poor socioeconomic status and lesser IQ of the mother, were predictors for poor cognitive performance at 9 years in children. Also, our study showed that recovery from stunting at later years in children, who were stunted at 2 years, also to be a factor for better cognitive performance at 9 years. Thus, we would summarize that cognition at 9 years is because of interplay of multiple factors during the childhood period, which is inclusive of nutrition, timing of catch up growth, socioeconomic status and caregiver’s IQ. Changes have been made in the conclusion. 20, lines – 404-406

11.Do the authors think that catch up growth could indicate a recovery from earlier malnutrition leading to a similar catch up in cognition? Or that those who catch up avoid falling further behind, while those who remain stunted continue to fall behind? Their results could support either argument, but which argument is correct has important policy implications and this should be discussed.

 Findings from our study support the evidence that children who catch up perform better in cognition tests at 9 years, when compared to children, who have been persistently stunted between 2 and 9 years. This has been discussed in the lines 350-371 -

11 The authors from time to time slip into causal language, this should be avoided.

 Thank you and this has been addressed -

12. The discussion on the importance of the first 1,000 days is a little unclear and could be improved. It is the pace and foundational nature of development in that period which is critical and this does not come through enough.

 Thank you and this has now been addressed. 19, lines 364-371, & 

20, 391-393

13. Why report weight if you don't use it? Just for context?

 Yes, weight for age scores have been provided to give a complete picture of anthropometric measurements of children in the cohort. -

14. The top third based on WAMI can hardly be called high SES given the selection process. Would relatively high not be better?

 Thank you. We have used this nomenclature of high and low, having 33rd percentile as cut-off consistently in the previous papers published from the same cohort. Hence, we would like to retain the same nomenclature. 

Mohan VR, Sharma S, Ramanujam K, Babji S, Koshy B, Bondu JD, John SM, Kang G. Effects of elevated blood lead levels in preschool children in urban Vellore. Indian Pediatr. 2014 Aug;51(8):621-5.

Koshy B, Srinivasan M, Zachariah SM, Karthikeyan AS, Roshan R, Bose A, Mohan VR, John S, Ramanujam K, Muliyil J, Kang G. Body iron and lead status in early childhood and its effects on development and cognition: a longitudinal study from urban Vellore. Public Health Nutr. 2020 Aug;23(11):1896-1906. -

15. When was the WAMI measure taken? What the implications of only using one time point? Did those who exhibited catch up also see an improvement in their SES which might explain the better cognitive scores? WAMI measures were available for the MAL-ED study, until 5 years of age and there were no WAMI measurements, thereafter. In the current study, WAMI scores measured at 2 years was considered, since we were interested to see, if SES status post-infancy continued to have effect on cognition scores measured at later childhood. 

Further, subgroup analysis showed that there is no significant improvement in SES status between the children who were stunted at 2 years and recovered at 5 years vs. those who were stunted both at 2 and 5 years. To substantiate this, we would provide following data. Of 95 children, who were stunted at 2 years, 37 (38.9%) recovered from stunting, with 58 (61.1%) of them, continued to be stunted at 5 years. 4/37 (10.8%) children who recovered from stunting had the improvement in SES from low to high by 5 years; compared to 5/58 (8.62%) who continued to be stunted at 5 years and had a similar improvement in SES status from low to high. -

Reviewer 2 

16. Table 4: on the PIQ, it is shown that stunting at 2 y, though with a catch-up at 5 y had 5.77 PIQ less than normal children, but the children who stunted at 2 & 5 who only caught-up at 9 had much better PIQ than normal, than children who were already catch-up since 2y and those persisted till 9 y. What could explain such a pattern? Please add discussion Thank you. We are unable to substantiate this pattern of regression co-efficients obtained between stunting with/without catch-up in early childhood and PIQ measurements at 9 years. The probable reason would be smaller sample size within the subgroups, that could have limited meaningful inference of the data. Having a smaller sample size in this study has already been highlighted as one of the limitations within the Discussion section. 20, line 394

17. . Lines 365-366 & reference 17: Did children who received intervention were during older childhood (e.g., early school-age) and hence, the damage on cognition could have occurred since early age, aside from possible short duration of intervention? Please consider.

 Thank you. We have added this into the Discussion. 19, Line 383-384

18. 23. Editorial errors:

a. Line 116, Introduction: ‘…. (HAZ) <2 on WHO…’, should this read ‘…. (HAZ) <-2 SD…’?

b. Line 351 ‘A unit increase in HAZ <2 Y…’, suggest to read, ‘A unit increase in HAZ at <2 y of age..’ Thank you.

Corresponding changes made 

 a. Page 5; line 117

b. Page 18; line 361

Thanking you

Yours sincerely

Authors

---

## [Decision Letter · Decision Letter 1]

1 Dec 2021

PONE-D-21-21418R1Are early childhood stunting and catch-up growth associated with school age cognition? – Evidence from an Indian birth cohortPLOS ONE

Dear Dr. Koshy,

Thank you for submitting your manuscript to PLOS ONE. After careful consideration, we feel that it has merit but does not fully meet PLOS ONE’s publication criteria as it currently stands. Therefore, we invite you to submit a revised version of the manuscript that addresses the points raised during the review process.

Thanks for the revised manuscript. Please revise according to the reviewers' comments and return it to us. Please submit your revised manuscript by Jan 15 2022 11:59PM. If you will need more time than this to complete your revisions, please reply to this message or contact the journal office at plosone@plos.org. Please include the following items when submitting your revised manuscript:A rebuttal letter that responds to each point raised by the academic editor and reviewer(s). You should upload this letter as a separate file labeled 'Response to Reviewers'.A marked-up copy of your manuscript that highlights changes made to the original version. You should upload this as a separate file labeled 'Revised Manuscript with Track Changes'.An unmarked version of your revised paper without tracked changes. You should upload this as a separate file labeled 'Manuscript'.If applicable, we recommend that you deposit your laboratory protocols in protocols.io to enhance the reproducibility of your results. Protocols.io assigns your protocol its own identifier (DOI) so that it can be cited independently in the future. For instructions see: https://journals.plos.org/plosone/s/submission-guidelines#loc-laboratory-protocols. Additionally, PLOS ONE offers an option for publishing peer-reviewed Lab Protocol articles, which describe protocols hosted on protocols.io. Read more information on sharing protocols at https://plos.org/protocols?utm_medium=editorial-email&utm_source=authorletters&utm_campaign=protocols.

We look forward to receiving your revised manuscript.

Kind regards,

Seo Ah Hong, PhD

Academic Editor

PLOS ONE

Journal Requirements:

Reviewers' comments:

Reviewer's Responses to Questions

**Comments to the Author**

1. If the authors have adequately addressed your comments raised in a previous round of review and you feel that this manuscript is now acceptable for publication, you may indicate that here to bypass the “Comments to the Author” section, enter your conflict of interest statement in the “Confidential to Editor” section, and submit your "Accept" recommendation.

Reviewer #1: (No Response)

Reviewer #2: All comments have been addressed

2. Is the manuscript technically sound, and do the data support the conclusions?

Reviewer #1: Partly

Reviewer #2: Yes

3. Has the statistical analysis been performed appropriately and rigorously? 

Reviewer #1: Yes

Reviewer #2: Yes

4. Have the authors made all data underlying the findings in their manuscript fully available?

Reviewer #1: Yes

Reviewer #2: Yes

5. Is the manuscript presented in an intelligible fashion and written in standard English?

Reviewer #1: No

Reviewer #2: Yes

6. Review Comments to the Author

Reviewer #1: The authors have not substantially engage with a number of key issues raised in the previous review. Most notably, they have not justified their definition of catch up growth other than to say that other people use it, so it must be okay. They do not even mention what the other definitions are and how these could explain the contradictory findings that they also seek to explain.

They also do not engage with the possibility that their results are endogenous - i.e. growth and cognition both have common causes. This links to perhaps the most serious problem - the authors use causal language throughout. Growing taller will not make you smarter. Growing taller may indicate that your environment has improved and that this environment now better facilitates cognitive development. This mechanism is not explained or referenced when interpreting findings. The absence of this explanation (understanding?) may explain why the conclusions are over stated. It may also explain why the results from the literature discussed are also over interpreted as showing a causal link.

Finally, the authors ignored the need to differentiate between catch up in cognition and a slowing of the rate at which children are falling behind. If children who remain stunted are falling behind at a faster rate than those who have 'caught up', then a significant result in their regression may not be indicating catch up in cognition in the latter group. This is a critical point - it speaks to the extent to which inputs in different life stages are or are not substitutable.

This paper presents data from a high quality study. The analysis is sound. My concerns should be relatively easy to address, they relate only to the careful framing of the problem and results.

Reviewer #2: I have no further additional comment. The authors responded to all comments, point-by-point adequately.

7. PLOS authors have the option to publish the peer review history of their article (what does this mean?). If published, this will include your full peer review and any attached files.

Reviewer #1: No

Reviewer #2: No

---

## [Author Response · Author response to Decision Letter 1]

14 Dec 2021

Separate copy is uploaded

To

The reviewers

PLOS One

Re: PONE-D-21-21418 - Are early childhood stunting and catch-up growth associated with school age cognition? – Evidence from an Indian birth cohort

Dear reviewers, 

Thank you for reviewing our manuscript titled “Are early childhood stunting and catch-up growth associated with school age cognition? – Evidence from an Indian birth cohort”. Author’s responses are provided below

Comments Authors’ response Page number; line number

1. The authors have not substantially engage with a number of key issues raised in the previous review. Most notably, they have not justified their definition of catch up growth other than to say that other people use it, so it must be okay. They do not even mention what the other definitions are and how these could explain the contradictory findings that they also seek to explain. Thank you.

Different definitions have been included in the introduction – page 6, lines 126-128.

Justification for current approach is included in methodology – page 8, lines 163-166

 6; 126-128

8; 163-166

2. They also do not engage with the possibility that their results are endogenous - i.e. growth and cognition both have common causes. This links to perhaps the most serious problem - the authors use causal language throughout. Growing taller will not make you smarter. Growing taller may indicate that your environment has improved and that this environment now better facilitates cognitive development. This mechanism is not explained or referenced when interpreting findings. The absence of this explanation (understanding?) may explain why the conclusions are over stated. It may also explain why the results from the literature discussed are also over interpreted as showing a causal link. Thank you.

We completely agree with the lack of causalilty and have used the word ‘association’ throughout manuscript.

We are happy to edit/modify if this is not clear in the manuscript. 

Another statement also has been added to highlight common and intersecting risks and causative pathways for both growth and cognition in early childhood. 20; 360-362

3. Finally, the authors ignored the need to differentiate between catch up in cognition and a slowing of the rate at which children are falling behind. If children who remain stunted are falling behind at a faster rate than those who have 'caught up', then a significant result in their regression may not be indicating catch up in cognition in the latter group. This is a critical point - it speaks to the extent to which inputs in different life stages are or are not substitutable. Thank you.

That is a valid observation and we agree with you. 

We have added further information in Table 2 to show that children in all groups improved in HAZ between 2 and 9 years of age.

 Table 2

Thanking you

Yours sincerely

Authors

---

## [Decision Letter · Decision Letter 2]

23 Dec 2021

PONE-D-21-21418R2Are early childhood stunting and catch-up growth associated with school age cognition? – Evidence from an Indian birth cohortPLOS ONE

Dear Dr. Koshy,

Thank you for submitting your manuscript to PLOS ONE. After careful consideration, we feel that it has merit but does not fully meet PLOS ONE’s publication criteria as it currently stands. Therefore, we invite you to submit a revised version of the manuscript that addresses the points raised during the review process.

Thanks for the revision. You made efforts to improve the manuscript.

However, some points which the reviewer #1 pointed out are so significant. So I would like you to revise once again throughout the manuscript according to the reviewer’s comments.

Further, there are several miner things which the authors should correct.

Table 3 showed the scores among 4 categories of stunting and catch up status of children and the coefficient of the rest three group, compared to the “never stunted group” were shown in Table 4. However, the authors used causal languages as the reviewer #1 mentioned. Thus, the uses of “reduction” or “decline” are not appropriate. You may use “lower score” instead. Please revise throughout the manuscript.

Reg “Catch-up growth was associated with higher verbal intelligence as well total cognition, but not the performance component.” higher verbal intelligence as well as total cognition than what? you mean “never catch up group”? Please clarify the sentences.

The authors did not discuss why the second group (catch up at 5 years) are negatively associated with performance IQ. Make some efforts on it, and reduce the length of “good follow-up rate of your cohort study” and add after the limitation of your study” to highlight the significance of your study in the discussion.

Lastly, put the acronym in parentheses after the full term, if you use the term the first time. For example, IQ in the abstract, SD in the introduction and ANOVA in the method section.

We look forward to receiving your revised manuscript.

Kind regards,

Seo Ah Hong, PhD

Academic Editor

PLOS ONE

Journal Requirements:

Reviewers' comments:

Reviewer's Responses to Questions

**Comments to the Author**

1. If the authors have adequately addressed your comments raised in a previous round of review and you feel that this manuscript is now acceptable for publication, you may indicate that here to bypass the “Comments to the Author” section, enter your conflict of interest statement in the “Confidential to Editor” section, and submit your "Accept" recommendation.

Reviewer #1: All comments have been addressed

2. Is the manuscript technically sound, and do the data support the conclusions?

Reviewer #1: Partly

3. Has the statistical analysis been performed appropriately and rigorously? 

Reviewer #1: Yes

4. Have the authors made all data underlying the findings in their manuscript fully available?

Reviewer #1: Yes

5. Is the manuscript presented in an intelligible fashion and written in standard English?

Reviewer #1: No

6. Review Comments to the Author

Reviewer #1: The authors have made some effort to engage with my comments, but some issues remain. There are still come remnants of causal language, such as 'reduction' and 'decline', but they are not serious and could easily be left. The justification of the definition use is weak, it mentions only the advantages and fails to consider the limitations (of which there are many). But as mentioned, previously, the definition is often used and so I guess its unquestioned use here is acceptable.

The one remaining issue which I would like to see addressed relates to the failure to engage with the comment I made that there may be no catch up in cognition. The additions to the table to show that the z-score increased don't help and add little. The definitions of the categories already provided the outline of this information. Besides, the question was not about growth but cognition. Perhaps my explanations have not been clear, so I will try with an example.

Three people enter a race (Tom, Fred and Joann). At the half way mark, Tom and Fred are 50meters behind Joann. When Joann crosses the finish line, Fred is still 50meters behind, but Tom is now 100meters behind. Can you say that Fred caught up? They all moved forward and he stopped falling behind, but he did not catch up.

This is a critical point, because if that 50meters at half way is lost forever and all you can do post the half way mark is prevent someone falling further behind, interventions in the first half of the race are not substitutable. This does not mean that interventions in the second half to prevent further falling behind are not important, but there should not be any suggestion that we can wait to the second half and catch up all the way.

In my mind, the possibility that there was no catch up in cognition should be mentioned. And this should be carried through to the conclusions, i.e. that the consequences of early deprivation on cognition may not be reversible, but we can act to prevent children from falling further behind.

7. PLOS authors have the option to publish the peer review history of their article (what does this mean?). If published, this will include your full peer review and any attached files.

Reviewer #1: No

---

## [Author Response · Author response to Decision Letter 2]

27 Jan 2022

To

The reviewers

PLOS One

Re: PONE-D-21-21418 - Are early childhood stunting and catch-up growth associated with school age cognition? – Evidence from an Indian birth cohort

Dear editor and reviewers, 

Thank you for reviewing our manuscript titled “Are early childhood stunting and catch-up growth associated with school age cognition? – Evidence from an Indian birth cohort”. Author’s responses are provided below

Comments Authors’ response Page number; line number

(Track version of the manuscript)

Further, there are several miner things which the authors should correct.

Table 3 showed the scores among 4 categories of stunting and catch up status of children and the coefficient of the rest three group, compared to the “never stunted group” were shown in Table 4. However, the authors used causal languages as the reviewer #1 mentioned. Thank you for the comment and the category ‘never stunted’ in table 2, 3 and 4 is replaced with ‘children who were never stunted’ Table 2, 3 and 4

Thus, the uses of “reduction” or “decline” are not appropriate. You may use “lower score” instead. Please revise throughout the manuscript. Thank you and this correction has been made in the Abstract, Results and Conclusions section. Page 4 and lines 83 and 88

Page 15 and lines 284, 287 and 291

Page 22 and lines 406 - 410

Reg “Catch-up growth was associated with higher verbal intelligence as well total cognition, but not the performance component.” higher verbal intelligence as well as total cognition than what? you mean “never catch up group”? Please clarify the sentences Thank you and this sentence has been rephrased. 

 Page 18 and lines 318-319

The authors did not discuss why the second group (catch up at 5 years) are negatively associated with performance IQ. Thank you and this change has been made in the Discussion Page 21 and lines 392-395

Make some efforts on it, and reduce the length of “good follow-up rate of your cohort study” and add after the limitation of your study” to highlight the significance of your study in the discussion. Thank you and this change has been made in the Discussion Page 21 and lines 402-403

Lastly, put the acronym in parentheses after the full term, if you use the term the first time. For example, IQ in the abstract, SD in the introduction and ANOVA in the method section. Thank you and these suggestions have been incorporated in the Abstract, Introduction and Methods. Page 4 and line 84

Page 5 and line 103

Page 10 and lines 214-215 

There are still come remnants of causal language, such as 'reduction' and 'decline', but they are not serious and could easily be left Thank you and this correction has been made throughout the manuscript. Page 4 and lines 83 and 88

Page 15 and lines 284, 287 and 291

Page 20 and line 360

The justification of the definition use is weak, it mentions only the advantages and fails to consider the limitations (of which there are many). But as mentioned, previously, the definition is often used and so I guess its unquestioned use here is acceptable. Thank you and the justification for the use of definition based on HAZ scores in this study has been provided in Methods section. Page 8 and lines 170 - 173

The one remaining issue which I would like to see addressed relates to the failure to engage with the comment I made that there may be no catch up in cognition. The additions to the table to show that the z-score increased don't help and add little. The definitions of the categories already provided the outline of this information. Besides, the question was not about growth but cognition. Perhaps my explanations have not been clear, so I will try with an example.

Three people enter a race (Tom, Fred and Joann). At the half way mark, Tom and Fred are 50meters behind Joann. When Joann crosses the finish line, Fred is still 50meters behind, but Tom is now 100meters behind. Can you say that Fred caught up? They all moved forward and he stopped falling behind, but he did not catch up.

This is a critical point, because if that 50meters at half way is lost forever and all you can do post the half way mark is prevent someone falling further behind, interventions in the first half of the race are not substitutable. This does not mean that interventions in the second half to prevent further falling behind are not important, but there should not be any suggestion that we can wait to the second half and catch up all the way.

In my mind, the possibility that there was no catch up in cognition should be mentioned. And this should be carried through to the conclusions, i.e. that the consequences of early deprivation on cognition may not be reversible, but we can act to prevent children from falling further behind. Thank you and we agree with this suggestion and corresponding changes have been made throughout in the manuscript. Use of ‘Catch up in cognition’ in children who recovered from early life stunting has been replaced as the attainment of higher cognition scores in these children compared to persistently stunted children in relevant sections. Page 4 and lines 85-86, 89-91

Page 18 and lines 318-319

Page 21 and lines 385-389

Page 22 and line 408-410

Thanking you

Yours sincerely

Authors

---

## [Decision Letter · Decision Letter 3]

2 Feb 2022

Are early childhood stunting and catch-up growth associated with school age cognition? – Evidence from an Indian birth cohort

PONE-D-21-21418R3

Dear Dr. Koshy,

We’re pleased to inform you that your manuscript has been judged scientifically suitable for publication and will be formally accepted for publication once it meets all outstanding technical requirements.

Kind regards,

Seo Ah Hong, PhD

Academic Editor

PLOS ONE

Additional Editor Comments (optional):

Thank you for addressing all the issues reviewers raised. Copy editing is strongly recommended to improve readability before submitting the final version since some errors were found. For example, change from “height for age SD scores (HAZ) < -2 SD on WHO growth charts” to “height for age z scores (HAZ) < -2 Standard Deviation (SD) on WHO growth charts”

Reviewers' comments:

Reviewer's Responses to Questions

**Comments to the Author**

1. If the authors have adequately addressed your comments raised in a previous round of review and you feel that this manuscript is now acceptable for publication, you may indicate that here to bypass the “Comments to the Author” section, enter your conflict of interest statement in the “Confidential to Editor” section, and submit your "Accept" recommendation.

Reviewer #1: All comments have been addressed

2. Is the manuscript technically sound, and do the data support the conclusions?

Reviewer #1: Yes

3. Has the statistical analysis been performed appropriately and rigorously? 

Reviewer #1: Yes

4. Have the authors made all data underlying the findings in their manuscript fully available?

Reviewer #1: Yes

5. Is the manuscript presented in an intelligible fashion and written in standard English?

Reviewer #1: Yes

6. Review Comments to the Author

Reviewer #1: The changes made in this revision are very helpful. The causal language has been removed and the interpretation of the findings has been improved. The paper makes a useful contribution to the literature.

7. PLOS authors have the option to publish the peer review history of their article (what does this mean?). If published, this will include your full peer review and any attached files.

Reviewer #1: No

---

## [Editor Report · Acceptance letter]

8 Feb 2022

PONE-D-21-21418R3 

Are early childhood stunting and catch-up growth associated with school age cognition? – Evidence from an Indian birth cohort 

Dear Dr. Koshy:

I'm pleased to inform you that your manuscript has been deemed suitable for publication in PLOS ONE. Congratulations! Your manuscript is now with our production department. 

Kind regards, 

on behalf of

Prof. Seo Ah Hong 

Academic Editor

PLOS ONE